# Enhancement of Hydrophilicity of Nano-Pitted TiO_2_ Surface Using Phosphoric Acid Etching

**DOI:** 10.3390/nano13030511

**Published:** 2023-01-27

**Authors:** Ferenc Koppány, Krisztián Benedek Csomó, Edvárd Márton Varmuzsa, Eszter Bognár, Liza Pelyhe, Péter Nagy, Imre Kientzl, Dániel Szabó, Miklós Weszl, Gábor Dobos, Sándor Lenk, Gábor Erdei, Gábor Kiss, Lilien Nagy, Attila Sréter, Andrea Alexandra Belik, Zsuzsanna Tóth, János Vág, Árpád Joób-Fancsaly, Zsolt Németh

**Affiliations:** 1Department of Oro-Maxillofacial Surgery and Stomatology, Semmelweis University, 1085 Budapest, Hungary; 2FERR-VÁZ Kft, 1161 Budapest, Hungary; 3Department of Translational Sciences, Semmelweis University, 1085 Budapest, Hungary; 4Department of Atomic Physics, Institute of Physics, Budapest University of Technology and Economics, 1111 Budapest, Hungary; 5Department of Prosthodontics, Semmelweis University, 1088 Budapest, Hungary; 6Faculty of Dentistry, Semmelweis University, 1085 Budapest, Hungary; 7Department of Molecular Biology, Semmelweis University, 1094 Budapest, Hungary; 8Department of Restorative Dentistry and Endodontics, Semmelweis University, 1088 Budapest, Hungary

**Keywords:** phosphoric acid, hydrophilicity, nano, titanium-dioxide, implant

## Abstract

Our research group developed a novel nano-pitted (NP) TiO_2_ surface on grade 2 titanium that showed good mechanical, osteogenic, and antibacterial properties; however, it showed weak hydrophilicity. Our objective was to develop a surface treatment method to enhance the hydrophilicity of the NP TiO_2_ surface without the destruction of the nano-topography. The effects of dilute and concentrated orthophosphoric (H_3_PO_4_) and nitric acids were investigated on wettability using contact angle measurement. Optical profilometry and atomic force microscopy were used for surface roughness measurement. The chemical composition of the TiO_2_ surface and the oxidation state of Ti was investigated using X-ray photoelectron spectroscopy. The ccH_3_PO_4_ treatment significantly increased the wettability of the NP TiO_2_ surfaces (30°) compared to the untreated control (88°). The quantity of the absorbed phosphorus significantly increased following ccH_3_PO_4_ treatment compared to the control and caused the oxidation state of titanium to decrease (Ti^4+^ → Ti^3+^). Owing to its simplicity and robustness the presented surface treatment method may be utilized in the industrial-scale manufacturing of titanium implants.

## 1. Introduction

Surface roughness and hydrophilicity have long been known as two key properties that promote the osseointegration of titanium implants [1]. This idea is supported by a wide range of high-quality scientific data that revealed the correlation between either micro- or nano- roughness or wettability and the biological properties of titanium-dioxide (TiO_2_) surfaces [2]. However, the transferability of the scientific achievements from experimental settings into industrial-scale manufacturing is often limited by the low productivity of the underlying technologies. Our research group has previously developed an electrochemical surface treatment method that enables the reproducible production of a novel nano-pitted (NP) TiO_2_ surface on grade 2 titanium discs and dental implants that showed good mechanical, osteogenic, and antibacterial properties [3,4]. 

Various methods have been proposed to enhance the hydrophilicity of titanium implants, including but not limited to UV irradiation, gamma irradiation, and thermal treatment. Each of these methods have disadvantages that detrimentally affect their applicability in the industrial-scale manufacturing of implants. For example, several studies have been published touting the benefits of UV irradiation that temporary turned dental implant surfaces to super hydrophilic, albeit they returned to a hydrophobic state in the dark [5]. Ueno et al. argues in favor of gamma-irradiation to regain the hydrophilicity and bioactivity of titanium implants that are lost over time owing to hydrocarbon deposition from the atmosphere [6]. However, a 25–35 kGy dose of gamma irradiation, which is generally used for the sterilization of titanium implants, might cause irreversible alteration to the TiO_2_ nanosurfaces as has been the case in other nanoparticles of various materials [7,8]. Thermal treatment can also temporarily increase the hydrophilicity of titanium implants, but the effect often decreases over time, and the heat may modify the surface morphology [9,10].

Etching may offer a simple and adaptable method to modify the wettability of TiO_2_ surfaces [11]. Recently, phosphoric acid has raised a great deal of interest because of its ability to improve the bioactivity of titanium implants. The phosphorus functionalization of TiO_2_ surfaces has been proven to be a potent approach to enhance both the catalytic and the biological properties of titanium [12,13]. However, the current methods of phosphorus-functionalization often apply concomitant exposure of phosphoric acid and physical energy (such as thermal or electrochemical treatment) to the TiO_2_ surface, which may irreversibly change the surface morphology [14].

Our objective was to find a cost-effective and adaptable surface treatment method that is suitable to permanently enhance the hydrophilicity of the TiO_2_ surface without destructing the nano-topography. Therefore, we investigated the effect of various etchants on the NP surface that was developed earlier by our research team.

## 2. Materials and Methods

### 2.1. Preparation of Titanium Discs

Grade 2 titanium rods were cut to discs (2 mm thick and 14 mm in diameter) using lathe turning. The discs were cleaned in a two-step process. In the first step, they were subjected to ultrasonic cleaning in acetone (Molar Chemicals, Halásztelek, Hungary) for 5 min at room temperature and then they were dried at room temperature. In the second cleaning step, the discs were washed in the ultrasonic cleaner in absolute ethanol (Molar Chemicals, Hungary) for 5 min at room temperature and then left to dry at room temperature.

### 2.2. Preparation of Nano-Pitted (NP) TiO_2_ Surfaces

Nano-pitted (NP) TiO_2_ surfaces were created using a protocol that was published previously by certain members of our research group [3]. Following this protocol, the titanium discs were briefly subjected to a three-step surface treatment process. In the first step, the machining marks were removed using electrochemical polishing in a two-electrode setup using a DC power source (Elektro-Automatik, EA-PS8080-40) in a steady electrolyte (NANOTI EP Electrolyte, NANOTI Ltd., Sutton Coldfield, UK) flow with 0.1 L/min velocity generated using a thermoplastic mag drive centrifugal pump (HTM6 PP, GemmeCotti), while the temperature of the electrolyte was kept at 15 °C applying 30 V for 35 s. In the second step, the polished discs were subjected to acid etching in the compound of 0.1 wt% HF, 1 wt% H_3_PO_4_, and distilled water (Molar Chemicals, Hungary) in an ultrasonic bath for 3 min at room temperature. In the third step, the NP anodic film was grown on the surface of the discs in a two-step anodic oxidation process in a two-electrode electrochemical reactor applying a continuous direct power supply (Elektro-Automatik, EA-PS 8360–15 2 U). The first step was carried out in hydrofluoric acid at 20 V (DC) voltage for 3 min. The second step was carried out in hydrochloric acid at 14 V (DC) voltage for 1 min.

### 2.3. Preparation of Samples

The discs with NP TiO_2_ surface were further processed by applying various etchants. The etching parameters were identical for each workpiece except for the applied etchants (Table 1).

### 2.4. Experimental Groups

Five experimental groups (one control and four test groups) were created to investigate the effect of various etchants on the physicochemical properties of NP surfaces. As a control, untreated NP surfaces were used in the study. In each experimental group, one sample was tested. On each sample three regions of interest were randomly selected where the contact angle and optical measurements were carried out. The robustness of the surface treatment method justified the low sample number in the experiment [3].

### 2.5. Optical Profilometry

The surface roughness of the 5 experimental groups was measured by Optical Profilometry (Bruker Contour GT-K0X). An 800 × 600 μm^2^ size region of interest (ROI) was randomly selected on each sample. The arithmetical mean height (Ra) and root mean square deviation (Rq) values were measured on the samples.

### 2.6. Contact Angle Measurement

Distilled water was used as test fluid for contact angle measurement on the surface of the samples using a drop shape analyzer (Krüss DSA25, Hamburg, Germany). The measurement started immediately seconds after dropping, the drop volume was 3 ± 0.3 µL. The Elipse (Tangent-1) fitting method (Advance software, Krüss, Hamburg, Germany) was used to determine the left and right contact angles of the drop. One drop was added per sample and the contact angle of this drop was determined by 35 measurements. The mean of these measurements characterized the contact angle of the sample.

### 2.7. Atomic Force Microscopy

Atomic force microscopy (AFM) (Bruker Dimension Icon) was used to confirm the results of the optical profilometry. AFM measurements were performed in tapping mode using a Tap300Al g tip. The measurements were performed on the surface areas where obvious surface flaws did not appear. The measurements were performed alongside the full diameters of the discs. The arithmetical mean height (Ra) and root mean square deviation (Rq) values were measured on the samples.

### 2.8. X-ray Photoelectron Spectroscopy

The reference (NP) and the most hydrophilic samples were also studied using X-ray Photoelectron Spectroscopy (XPS). The XPS instrument used in the study was a custom-built system. The spectra were recorded using Mg Kα radiation from a Thermo Fisher XR4 dual anode X-ray source and a Specs Phoibos 150 hemispherical energy analyzer was used to measure the energy distribution of the photoelectrons. The background pressure in the analytical chamber was 2 × 10^−9^ mbar, but it rose to 4 × 10^−9^ mbar during the measurements due to the degassing of the samples. Due to the oxide coverage on the surface, a small amount of charging was also observed. This was corrected based on the position of the adventitious carbon peak. After recording the surface spectra, each sample was subjected to 3 keV Ar^+^ ion-beam sputtering from a custom-built ion source for 10 min. The sputtering speed (calibrated on SiO_2_) was 30 nm/h. This removed roughly 5 nm from the surface of each sample. Subsequently, a set of new XPS spectra were recorded to compare the composition of the surface to deeper layers.

### 2.9. Statistical Analysis

The mean, standard deviation, and coefficient of variation were used from the descriptive statistics. The coefficient of variation (CV) is the normalized standard deviation (standard deviation divided by the mean). The CV value under 10% means a homogenous dataset, between 10 and 20% means a low heterogenous dataset, and between 20 and 30% means a very heterogenous dataset. Above 30%, the dataset is very volatile.

A Kruskal–Wallis test was performed to compare the samples (*p* < 0.05) and a Games-Howell test for the post hoc comparison. The Games-Howell test was deliberately selected for post hoc analysis because it was nonparametric, unlike the Tukey’s test, and it did not assume equal sample sizes and homogeneity of variances.

## 3. Results

The results showed that the treatment with various etchants did not cause any apparent changes to the NP surface in terms of surface roughness (Table 2).

The treatment of the NP samples with ccH_3_PO_4_ significantly reduced (*p* = 0.00) the contact angle compared to the untreated control or, in other words, increased the wettability of the NP surface (Figure 1). The coefficient of variance was less than 14%, meaning that the samples were either homogenous or low heterogenous; therefore, the wettability of the samples could be described using the mean of the measured values of the contact angle (Table 3).

From this point forward, only the untreated control NP and the ccH_3_PO_4_-treated NP samples were subjected to further experiments. The AFM measurement confirmed the results of the optical profilometry, i.e., the surface roughness of the NP samples remained unchanged after ccH_3_PO_4_ treatment (Table 4).

Analyzing the surface element composition revealed approximately ten times more phosphorus on the ccH_3_PO_4_-treated sample (before sputtering: 7.5%; after sputtering 9.1%) compared to the untreated control NP surface (before sputtering: 0.6%; after sputtering: 0.9%) (Table 5). The proportion of O to Ti was 7.4 on the ccH_3_PO_4_-treated sample before sputtering and it dropped to 3.8 after sputtering. The proportion of O to Ti was 10.3 on the untreated control NP sample before sputtering and it dropped to 3.1 after sputtering. Interestingly, 41.5% less carbon appeared on the surface of the ccH_3_PO_4_-treated NP surface than on the untreated control (Figure 2).

The XPS spectrum of the untreated control NP surface showed that the Ti 2*p*_3/2_ peak appeared at 459.1 eV binding energy indicating fully oxidized titanium (Ti^4+^) in the TiO_2_ oxide layer. When analyzing the ccH_3_PO_4_-treated NP surface, an additional peak appeared on the spectrum at 458.8 eV binding energy beside the peak at 459.7 eV, thus indicating the presence of less oxidized titanium (Ti^3+^) beside the Ti^4+^ in the TiO_2_ layer (Figure 3) [15].

## 4. Discussion

Our results showed that the concentrated phosphoric acid treatment significantly enhanced the hydrophilicity of the NP surface, while dilute phosphoric acid and nitric acid did not have a considerable effect on it. The only known explanation of the asymmetric peak broadening on the right-hand side of the XPS peak (Figure 3) is the presence of another chemical state, which corresponds to Ti^3+^ suggesting that the concentrated phosphoric acid treatment caused the decrease in the oxidation state of titanium (Ti4+ → Ti3+) in the oxide layer. The etching did not alter the surface roughness of the NP samples, thus indicating the good resistance of the NP topography to chemical exposure. Since the antibacterial effect of the NP surface is related to the texture distribution of its nanostructure, we can assume that it retains a good portion of this property after the treatment. The proportion of absorbed phosphorus significantly increased after concentrated phosphoric acid treatment of the NP surface compared to the untreated control. Interestingly, the proportion of carbon was more than 40% less on the phosphoric acid-treated NP surface than on the control. Our earlier and current results show that the NP surface is very resistant both mechanically and chemically, which causes it to be a promising candidate for the surface treatment of titanium bone implants. The resistance to chemical exposure is relevant because it allows the functionalization of the NP surface either with chemical or biological compounds to enhance the bioactivity of implants.

It was surprising how much a simple etching of the NP surface in the concentrated phosphoric acid increased the hydrophilicity. The current methods apply physical exposure beside phosphoric acid etching to perform the phosphorus functionalization of the TiO_2_ surfaces. Hence, existing theories to characterize the physicochemical nature of phosphorus adsorption or its effect on hydrophilicity may not be fully applicable to explain our findings [16]. We assume that the concentrated phosphoric acid was strong enough to penetrate the uppermost layer of TiO_2_ where the phosphate ions chemically reacted with the titanium. Based on the available data, it is not possible to explain the entire reaction mechanism, however the appearance of the 2*p*_3/2_ peak at 458.8 eV binding energy on the XPS spectra of the concentrated phosphoric acid-treated NP surface suggests that a redox reaction ocurred on the surface.

The oxidation state of the adhered phosphorus is not known. However, the presence of three times more calcium on the surface of the concentrated phosphoric acid-treated NP surface suggests that the phosphorus is coordinated with oxygen (O^−^) that can bind cations, such as calcium. Calcium is ubiquitous in the human body, in blood and interstitial fluids, meaning that the implant surface is exposed to it after surgical insertion. The coordination of phosphorous with calcium ions might increase the quantity and volume of charges on the titanium surface [17]. The appearance of the surface charge might be responsible for the raise in hydrophilicity of the NP surface after phosphoric acid treatment. There are two possible underlying mechanisms: (i) the presence of surface charges that can facilitate the adsorption of more water molecules than onto uncharged the TiO_2_ surface and (ii) the surface charge seemingly prevented the adhesion of hydrocarbons from the ambient air that were known to be responsible for the deterioration of the hydrophilicity of the TiO_2_ surfaces.

In conclusion, the etching of the nano-pitted TiO_2_ surface with ccH_3_PO_4_ improved its wettability and did not destroy the nano-topography.

The results of our study should be interpreted by taking into account its limitations. Geometry, such as the dental implant shape, may influence the homogeneity of the effect of phosphoric acid treatment across the surface, which raises the question of the adaptability of the applied surface-treatment method. In addition, in industrial-scale manufacturing, for the control of treatment parameters, temperature is of great importance to assure uniformity of end-products, for instance. The issues that may arise in industrial-scale manufacturing could not be modeled in our laboratory-scale experimental setting. The corrosion and mechanical resistance of the ccH_3_PO_4_-treated NP TiO_2_ surface needs to be investigated before a conclusion can be drawn regarding the industrial applicability of the surface-treatment method.

## 5. Patents

There are patent applications that cover the subject matter of the anodic surface treatment of titanium implants (PCT/IB2016/050464) and (P1600046) and patent pending P200019 (2021_SZABL_10_NANOPITTED).

## Figures and Tables

**Figure 1 nanomaterials-13-00511-f001:**
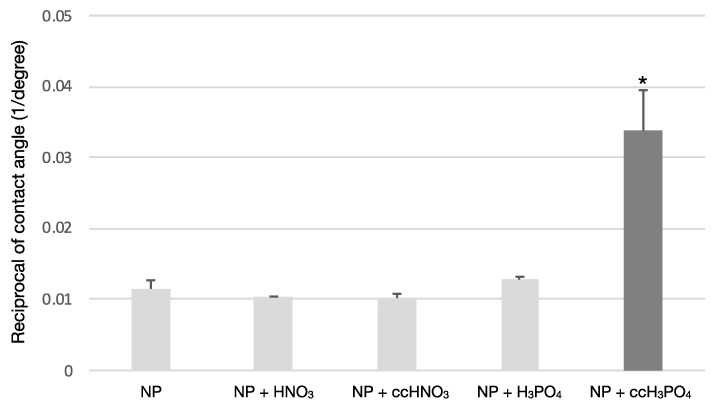
**The effect of surface treatments on the wettability of NP surface.** The treatment of the NP surface using ccH_3_PO_4_ has significantly increased the wettability compared to the untreated control (*p ** = 0.00). The other etchants did not cause an apparent effect on the wettability of the NP surface.

**Figure 2 nanomaterials-13-00511-f002:**
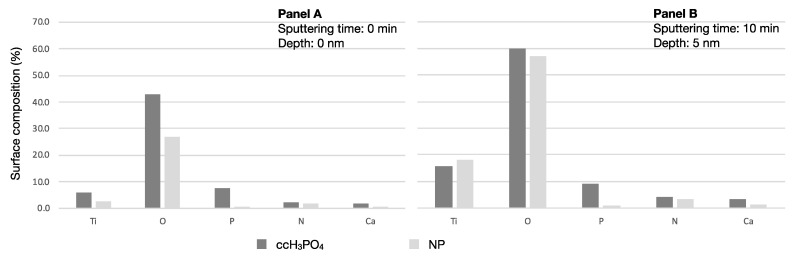
**Proportion of elements on the untreated control and the ccH_3_PO_4_-treated NP surfaces.** Panel A shows that the concentrations of phosphorus and oxygen are higher on the ccH_3_PO_4_-treated NP surface than on the untreated control. Panel B shows that the proportion of phosphorus did not change on the control or on the ccH_3_PO_4_-treated NP surfaces after sputtering. Carbon concentration is intentionally omitted because it would have caused the visualization of the other elements to be difficult (see Table 3 for data).

**Figure 3 nanomaterials-13-00511-f003:**
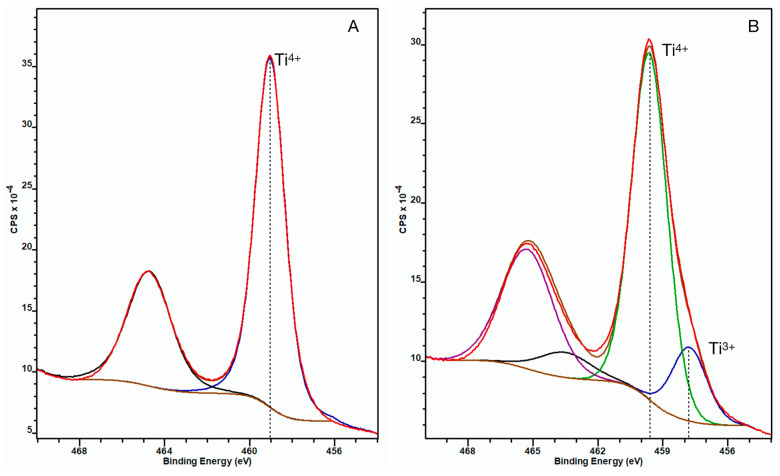
**XPS spectra of untreated control (A) and ccH_3_PO_4_-treated NP surfaces (B) after sputtering.** The Ti 2*p*_3/2_ peak appeared at 459.1 eV (blue curve) for the untreated NP surface, indicating fully coordinated Ti^4+^ ions, suggesting the oxide layer was solely constituted of TiO_2_. (The black curve is the corresponding Ti 2*p*_1/2_ component of the doublet peak.) On panel B, an additional doublet peak appeared at 458.8 eV binding energy, which indicates the presence of Ti^3+^ beside Ti^4+^ in the oxide layer. (Here the blue curve is Ti^3+^ 2*p*_3/2_, the green one is Ti^4+^ 2*p*_3/2_, while the black and purple curves are the corresponding Ti^3+^ 2*p*_1*/*_2 and Ti^4+^ 2*p*_1/2_ components of the doublets).

**Table 1 nanomaterials-13-00511-t001:** **The process of sample preparation.** Altogether, four test groups were prepared by etching the NP surfaces using various acids.

Workpiece	NP TiO_2_ Disc			
Etchant	Dilute orthophosphoric acid	Concentrated orthophosphoric acid	Dilute nitric acid	Concentrated nitric acid
Etchant composition	(1 g 85 wt% H_3_PO_4_ + 98.9 g dist. H_2_O)	(85 wt% H_3_PO_4_)	(1 g 65 wt% HNO_3_ + 98.9 g dist. H_2_O)	(65 wt% HNO_3_)
Test group name	NP + H_3_PO_4_	NP + ccH_3_PO_4_	NP + HNO_3_	NP + ccHNO_3_
Etching time	3 min			
Etching temperature	20–60 °C			
Agitation	None.			
Cleaning Step 1 Step 2 Step 3	Rinsing in ultrasonic bath: In distilled for 4 min at room temperature. In acetone for 5 min at room temperature. In absolute ethanol for 5 min at room temperature.			
Storage of samples	In hermetically sealed containers			

**Table 2 nanomaterials-13-00511-t002:** **Surface roughness of samples measured using optical profilometry.** Ra: arithmetical mean height indicates the average of the absolute value along the sampling length; Rq: root mean square deviation indicates the root mean square along the sampling length. Regarding Ra, the surface roughness of the of samples remained in the range from 126 nm to 196 nm with an average of 152.2 nm (±26.7 nm) and a median of 149 nm. Regarding Rq, the surface roughness values of the of samples ranged from 166nm to 249 nm with an average of 196.6nm (±31.9 nm) and a median of 191 nm.

Treatments	Surface Roughness	
	Ra (nm)	Rq (nm)
NP	149	191
NP + HNO_3_	153	199
NP + ccHNO_3_	126	166
NP + H_3_PO_4_	137	178
NP + ccH_3_PO_4_	196	249
Average Standard deviation Median	152.2 26.7 149	196.6 31.9 191

**Table 3 nanomaterials-13-00511-t003:** **Results of contact angle measurement.** The ccH_3_PO_4_ treatment of NP surface significantly reduced the contact angle (*p ** = 0.00). The reciprocal of the mean and standard deviation (SD) values concerning contact angle were calculated for a better visualization of the results in Figure 1.

Treatments	Contact Angle (°)			Reciprocal of Contact Angle (1/Degree)	
	Mean	SD	CV	Mean	SD
tgeNP	88.0	9.3	10.6	0.0115	0.0012
NP + HNO_3_	97.0	1.4	1.4	0.0103	0.0001
NP + ccHNO_3_	99.0	6.1	6.2	0.0102	0.0006
NP + H_3_PO_4_	78.0	2.5	3.2	0.0128	0.0004
NP + ccH_3_PO_4_	30.0 *	4.1	13.7	0.0338	0.0057

**Table 4 nanomaterials-13-00511-t004:** **Surface roughness measured using AFM.** Three regions of interest (ROI) were randomly selected on the untreated control and two on the ccH_3_PO_4_-treated NP surface for analysis. The treatment of NP surface with ccH_3_PO_4_ did not affect the surface roughness apparently.

Treatments	Surface Roughness	
	Ra (nm)	Rq (nm)
NP (ROI 1)	36	45
NP (ROI 2)	43.4	61.1
NP (ROI 3)	37.6	46.7
NP + ccH_3_PO_4_ (ROI 1)	45.7	71.8
NP + ccH_3_PO_4_ (ROI 2)	39.1	49.3

**Table 5 nanomaterials-13-00511-t005:** **Element composition of untreated control and ccH_3_PO_4_-treated NP surfaces.** Ten times more phosphorus was detected on the ccH_3_PO_4_-treated NP surface than on the control before and after sputtering. Approximately three times more calcium appeared on the ccH_3_PO_4_-treated NP surface than on the untreated control even after sputtering. Carbon and nitrogen measurements were presumably contaminations from the ambient air.

Sample Name	Sputtering [min]	Depth [nm]	C	Ti	O	P	N	Ca
NP + ccH_3_PO_4_	0	0	38.7	5.8	43.0	7.5	2.4	2.0
NP + ccH_3_PO_4_	10	5	6.0	15.7	60.0	9.1	4.2	3.5
NP	0	0	66.1	2.6	26.9	0.6	1.9	0.5
NP	10	5	18.9	18.3	56.9	0.9	3.5	1.5

## Data Availability

The data presented in this study are available on request from the corresponding author. The data are not publicly available due to patent pending.

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
