# Peer review of "Enhancement of Hydrophilicity of Nano-Pitted TiO2 Surface Using Phosphoric Acid Etching"

_nanomaterials, 2023, doi:10.3390/nano13030511_

Round 1

Reviewer 1 Report

This study compared several surface treatments of nanoscopic TiO2 surfaces that are important in dental materials.  The "NP" or native material shows only Ti(+4) on the surface.  The surfaces treated with phosphoric acid show higher hydrophilicity.  It has been known since the early days of metal treatment and metal-finishing that certain oxophilic metals (notably steel) can be phosphatized by such treatments, which renders them more receptive toward coatings such as paint, so it should not be a surprise that they are more hydrophilic.  A RAIRS infrared spectrum might help identify the relevant surface species.  The authors claim that surface treatments result in reduction of Ti(+4) to Ti(+3).  The basis of this claim is an XPS spectrum (Figure 3) that shows a very slight shoulder that can only be interpreted as an additional species by fairly creative applications of curve-fitting.  There isn't any chemistry that the authors were able to cite that would bring about redox chemistry with phosphate and Ti(+4), and the authors are not able to characterize the oxidation state of phosphorus in their samples anyway.  On the other hand, it is noteworthy that the very last step in the surface treatment is sonication with ethanol.  Ethanol is a known reducing agent for many high-oxidation-number transition metal ions.   I don't know about Ti(IV) but it's worth looking into that.  Only the NP surface was not sonicated with ethanol. It remains that the XPS evidence for the redox chemistry is very weak, and the rest of the surface characterization (aside from the contact angle measurements) was relatively unformative, so the analysis and conclusions drawn regarding the surface chemistry is mostly speculative.  As such I really can't support the publication of this article in this journal without better data.

Reviewer 2 Report

The paper can be considered for publication after Major revision.

Please see my attached file

Round 2

Reviewer 1 Report

Thank you for your response to my comments.  Good luck with your future work.

Reviewer 2 Report

Dear Authors,

Thanks for answering my questions. After reading the reply and the revised manuscript I think the paper is suitable for publication.

With regard to the AFM  figures I strongly suggest to add the figures in the text. In your case the AFM tip was sufficient for your investigation, However due to the fact that this is a common error,  the pictures eliminates any doubts